# Is It Possible to Create Antimicrobial Peptides Based on the Amyloidogenic Sequence of Ribosomal S1 Protein of *P. aeruginosa*?

**DOI:** 10.3390/ijms22189776

**Published:** 2021-09-10

**Authors:** Sergei Y. Grishin, Pavel A. Domnin, Sergey V. Kravchenko, Viacheslav N. Azev, Leila G. Mustaeva, Elena Y. Gorbunova, Margarita I. Kobyakova, Alexey K. Surin, Maria A. Makarova, Stanislav R. Kurpe, Roman S. Fadeev, Alexey S. Vasilchenko, Victoria V. Firstova, Svetlana A. Ermolaeva, Oxana V. Galzitskaya

**Affiliations:** 1Institute of Protein Research, Russian Academy of Sciences, 142290 Pushchino, Russia; syugrishin@gmail.com (S.Y.G.); alan@vega.protres.ru (A.K.S.); st.kurpe@gmail.com (S.R.K.); 2Gamaleya Research Centre of Epidemiology and Microbiology, 123098 Moscow, Russia; paveldomnin6@gmail.com (P.A.D.); drermolaeva@mail.ru (S.A.E.); 3Biology Faculty, Lomonosov Moscow State University, 119991 Moscow, Russia; 4Institute of Environmental and Agricultural Biology (X-BIO), Tyumen State University, 625003 Tyumen, Russia; svkraft@yandex.ru (S.V.K.); avasilchenko@gmail.com (A.S.V.); 5The Branch of the Institute of Bioorganic Chemistry, Russian Academy of Sciences, 142290 Pushchino, Russia; viatcheslav.azev@bibch.ru (V.N.A.); mustaeva@rambler.ru (L.G.M.); eyugorbunova@rambler.ru (E.Y.G.); 6Institute of Theoretical and Experimental Biophysics, Russian Academy of Sciences, 142290 Pushchino, Russia; kobyakovami@gmail.com (M.I.K.); fadeevrs@gmail.com (R.S.F.); 7State Research Center for Applied Microbiology and Biotechnology, 142279 Obolensk, Russia; mari.makar20@gmail.com (M.A.M.); victoria1@mail.ru (V.V.F.)

**Keywords:** ribosomal S1 protein, amyloid, antimicrobial peptides, *Pseudomonas aeruginosa*, cell penetrating peptide

## Abstract

The development and testing of new antimicrobial peptides (AMPs) represent an important milestone toward the development of new antimicrobial drugs that can inhibit the growth of pathogens and multidrug-resistant microorganisms such as *Pseudomonas aeruginosa,* Gram-negative bacteria. Most AMPs achieve these goals through mechanisms that disrupt the normal permeability of the cell membrane, which ultimately leads to the death of the pathogenic cell. Here, we developed a unique combination of a membrane penetrating peptide and peptides prone to amyloidogenesis to create hybrid peptide: “cell penetrating peptide + linker + amyloidogenic peptide”. We evaluated the antimicrobial effects of two peptides that were developed from sequences with different propensities for amyloid formation. Among the two hybrid peptides, one was found with antibacterial activity comparable to antibiotic gentamicin sulfate. Our peptides showed no toxicity to eukaryotic cells. In addition, we evaluated the effect on the antimicrobial properties of amino acid substitutions in the non-amyloidogenic region of peptides. We compared the results with data on the predicted secondary structure, hydrophobicity, and antimicrobial properties of the original and modified peptides. In conclusion, our study demonstrates the promise of hybrid peptides based on amyloidogenic regions of the ribosomal S1 protein for the development of new antimicrobial drugs against *P. aeruginosa*.

## 1. Introduction

Currently, the development of new antimicrobial peptides (AMPs) is an important area for bioengineering and biomedical applications [1,2,3]. AMPs are created as molecules that selectively affect the basic bacterial functions or growth processes of pathogenic organisms. Over the past decade, several AMPs have been developed that can effectively act on pathogenic bacteria, fungi, or viruses [4,5,6].

New technologies have been demonstrated for the development of new antimicrobial peptides based on predicting candidate amino acid sequences in proteins, as implemented in Webservers as AmpGram [7], AMP Scanner [8], and CAMPR3 [9]. The task of developing new approaches to counteracting the spread of the pathogen is especially important for bacterial cells, such as *P. aeruginosa*, which play a role in the development of serious diseases and adaptively respond to many drugs, acquiring resistance to antibiotics [10,11,12,13]. Various methods for overcoming the problem of bacterial resistance are presented in the work of Shemyakin et al. [14]. The adaptive resistance of *P. aeruginosa* includes biofilm formation as a diffusion barrier restricting the access of antibiotics to bacterial cells [15,16]. In addition, the main resistance mechanisms include the low permeability of the outer membrane. OprF is the predominant porin of *P. aeruginosa* and is responsible for the nonspecific uptake of ions and saccharides, but has a low efficiency for antibiotic translocation [17,18]. Current therapeutic options for the treatment of *P. aeruginosa* include the use of various combinations of antibiotics and the development of new antibiotics and antimicrobial peptides, as well as their combinations [19,20]. The problem is that pathogenic microorganisms often adapt to the action of drugs, for example, synthesizing AAC(6′)-I-type acetyltransferases, which catalyze the reaction of amikacin inactivation [21,22,23]. It has been shown that a wide range within efflux systems and mutations of *P. aeruginosa* strains evolutionarily increases the resistance of *P. aeruginosa* cells to antibiotics [24,25].

In this regard, amyloidogenic peptides are of interest, especially those that cause aggregation of ribosomal proteins and additionally disrupt the synthesis of bacterial proteins in general. Amyloidogenic peptides, alone or as part of combined preparations, can potentially be used to develop hybrid therapy for bacterial infections [26,27]. To achieve this goal, a deeper understanding of the mechanisms of action of peptides with amyloidogenic and antimicrobial properties is currently required [28,29,30]. The phenomenon of coaggregation of molecules, including those that act by the mechanism of directed coaggregation with a target lipid or protein, which is important for the function of antimicrobial peptides, is discussed in the recent scientific articles [31,32]. The mechanism of the transition of a protein from a native state to amyloid is not entirely clear. However, separate studies have examined bacterial amyloids [33]: amyloidogenesis of a prion-like protein was experimentally detected in *E. coli* using a fluorescent dye, and the RepA-WH1 protein formed cytotoxic aggregates that prevent bacterial growth [34]. The special attention should be paid to the use of the amyloidogenic properties of bacterial proteins for targeted protein aggregation [35]. The development of synthetic amyloidogenic peptides has been shown to be a promising approach to combat pathogenic bacteria [36,37]. At the same time, it should be borne in mind that bacteria have different mechanisms of counteracting protein aggregation, for example, by directing amyloids into inclusion bodies [35]. In addition, some bacteria are capable of using amyloids and amyloid-like protein structures in order to form biofilms [38,39]. Antimicrobial amyloidogenic peptides can be toxic not only to bacteria but also to eukaryotic cells. To avoid this, it is important to select a unique bacterial protein for induced amyloidosis in a bacterial cell and develop AMPs for this target.

We chose the ribosomal S1 protein as the target protein because this protein is unique to bacteria and is found as the largest bacterial ribosomal protein included in the 30S ribosome subunit. S1 protein performs many functions: it is required for the initiation of translation of bacterial proteins and regulation of translation [40]. Bioinformatic studies of the family of ribosomal S1 protein have recently demonstrated that the number of domains in S1 proteins is a characteristic of the phylogenetic grouping of bacteria and can vary from one to six [41]. Each S1 domain has a number of functions that are still not well known; nevertheless, the domains are very similar in structure and consist of an OB-fold [42]. OB-fold is a β-barrel including five antiparallel β-strands and a single α-helix. This fold is capable of performing a wide range of biological processes: control of RNA splicing, DNA repair, regulation of transcription, and many other functions [43,44]. The S1 domain can be found in different living organisms, and the number of S1 domains can vary from one to 15 [45]. The S1 protein from *P. aeruginosa* belongs to one of the largest classes of six-domain S1 proteins according to the server http://oka.protres.ru:4200 (accessed on 7 September 2021) [46].

Amyloidogenic regions within the S1 domains of ribosomal S1 protein from *P. aeruginosa* were originally predicted using FoldAmyloid [47], Waltz [48], AGGRESCAN [49], and Pasta 2.0 [50], after which it was experimentally demonstrated that there amyloidogenic regions forms amyloid fibrils in vitro [51].

The relationship between amyloidogenic and antimicrobial properties plays an important role in the development of hybrid AMPs. Thus, it is known that amyloidogenic peptides, like antimicrobial peptides, can have a toxic effect on bacterial cells [52]. In addition, the amino acid sequences included in the spine of amyloid fibrils are sufficiently resistant to the action of proteases, which is one of the diagnostic properties of amyloid fibrils [53,54]. This resistance can also be useful for antimicrobial peptides, allowing them to be protected from the action of exopeptidases [55]. Typically, such hybrid peptides entail reciprocal biophysical and biochemical interactions that depend on the amino acid sequence, its mobility and hydrophobicity. The amino acid sequence is believed to play a key role in achieving the correct structure and function of antimicrobial peptides [56]. However, amyloidogenic peptides can have a toxic effects on eukaryotic cells [57,58], which is an undesirable effect for the future use of AMPs in medicine [59]. Thus, the creation of a new generation of amyloidogenic-antimicrobial peptides requires studying how amyloidogenic and antimicrobial amino acid sequences can interact as part of a single molecule.

In accordance with the strategy presented in Figure 1, we developed hybrid peptides that contained a cell penetrating peptide (CPP) (fragment of Tat-peptide [60]) at the N-termini and an amyloidogenic sequence at the C-termini, which was predicted using bioinformatics tools [51]. The two sites were linked by four glycines. Thus, the construction of the fusion peptide was as follows: “CPP”+”linker GGGG”+ “predicted amyloidogenic-fragment”. The LHITDMAWKR and ITDFGIFIGL sequences were used as amyloidogenic fragments. Interestingly, although both of these sequences were predicted to be amyloidogenic, only the ITDFGIFIGL peptide exhibited amyloid-like properties according to electron microscopy and fluorescence of thioflavin T (ThT) under the condition studied [51]. In this paper, we tested the antimicrobial properties of the RKKRRQRRRGGGGLHITDMAWKR (R23R) and RKKRRQRRRGGGGITDFGIFIGL (R23L) peptides, as well as their analogs RKKRRQRRRGG-Sar-GLHITD-Nle-AWKR (R23R*) and RKKRRQRRRGG-Sar-GITDFGIFIGL (R23L*) to determine the possible effects associated with the replacement of glycine with sarcosine. In the R23R* peptide, methionine is replaced by norleucine, which is associated with the peculiarities of the R23R* peptide synthesis [61]. The antimicrobial properties of peptides can be seriously affected by a single point modification of a known sequence [62,63]. A similar approach based on the replacement of hydrophobic amino acid residues with sarcosine was used by other authors to show the importance of a stable conformation of the peptide for its antimicrobial action [64].

## 2. Results

### 2.1. Prediction of the Secondary Structure and Antimicrobial Propensities of the R23R, R23L, R23R*, and R23L* Peptides

The effect of amino acid substitutions on the secondary structure and some physicochemical properties of the peptides can be predicted using special programs [65,66,67]. Theoretical information on ordered elements of the secondary structure of peptides, physicochemical properties of R23R, R23L, R23R*, and R23L*presented in Table 1. We used the JPred4 (“jnetpred” algorithm) and PredictProtein (“RePROF” algorithm) programs to predict the elements of the secondary structure of the R23R, R23L, R23R*, and R23L* peptides. Considering that sarcosine is not used for calculations in the algorithms of these bioinformatics tools, this amino acid residue in the original peptide sequence was replaced by alanine (A) and proline (P) residues which are similar in structure and properties to take into account the changes in the prediction results caused by the substitution of glycine 12 to sarcosine.

As follows from the results of the predictions of the JPred4 and PredictProtein programs, substitution of position 12 glycine with sarcosine can lead to changes in the secondary structure of the modified R23R* and R23L* peptides. It was expected that replacing glycine with sarcosine would have a local effect on the secondary structure of the peptide, but this modification changed the results of predicting the secondary structure of the C-terminus of the peptide. According to the predictions we have a decrease in the β-structure of the modified peptides. At the same time, according to the data obtained using the DBAASP v3.0 database toolkit, the calculated µH_n_ and H_n_ values tend to decrease. Moreover, it follows from Table 1 that Z and I_0_ of the peptide do not change due to the modification caused by the replacement of 12 glycine to sarcosine.

The CAMPR3 [9], AmpGram [7], and AMP Scanner [8] Webservers, predicting the antimicrobial properties of peptides using their secondary structure and physicochemical properties, can be used to develop AMPs. To achieve this goal, we first of all predicted their antimicrobial properties (Table 2).

The replacement of glycine 12 with sarcosine did not significantly alter the predictions of antimicrobial properties by the CAMPR3, AmpGram, and AMP Scanner web servers. R23L and R23L* were predicted as antimicrobial peptides practically by all programs with exception CAMPR3 (SVM) and AmpGram. AMP properties were not predicted for R23R and R23R* using CAMPR3 (SVM) and AMP Scanner. As we see further these predictions does coincide with the experimental validation.

### 2.2. Experimental Validation of the Antibacterial Activity of Peptides

#### 2.2.1. Determination of the Antibacterial Activity of Peptides on Agar

In the next step, the predicted antimicrobial properties of the peptides were tested using agar experiments with *P. aeruginosa* strain PA103 cells (Figure 2).

The growth inhibition effects of *P. aeruginosa* cells were revealed for unmodified peptides R23R and R23L in the concentration ranges of 3600–3800 µM and 360–380 µM. Thus, for the R23R and R23L peptides, the value of the minimum inhibitory concentration (MIC) was 360 µM (1 mg/mL) and 380 µM (1 mg/mL), respectively. The modified peptides R23R* and R23L* practically do not give such an antibacterial effect as R23R and R23L (Figure 2A), that is, for them MIC ≥ 3600–3800 µM (10 mg/mL). At the same time, it should be borne in mind that the data were obtained for the *P. aeruginosa* strain PA103.

Interestingly, slightly different results on the antimicrobial properties of the peptides were found against the *P. aeruginosa* strain ATCC 28753 (Table 3).

Inhibition of the *P. aeruginosa* strain ATCC 28753 cell growth was detected only for R23L at a concentration of 300 µM (Table 3). The modified peptides have practically no antibacterial effect, for them the MIC > 300 µM. It should be noted that the differences in the antimicrobial effects of the R23R peptide in Figure 2 and in Table 3 are likely related to the different survival rates of strains PA103 and ATCC 28753 during co-incubation with this peptide.

In addition, the R23R, R23L, R23R*, and R23L* peptides were also tested on the *P. aeruginosa* strain ATCC 9027 (Table 4).

Comparing the sizes of the zones of inhibition of cell growth of the *P. aeruginosa* strain ATCC 9027, it was found that R23R, R23R*, and R23L* do not exhibit antimicrobial activity, while R23L exhibited activity against this strain at a high concentration of 3750 µM.

It should be noted that the RKKRRQRRRGG-Sar (R12-Sar) peptide itself did not exhibit antimicrobial activity at the concentrations used (see Appendix A).

#### 2.2.2. Measurement of the Antibacterial Activity of Peptides by Microdilution Technique

From the point of view of testing the antimicrobial properties of peptides, the growth conditions of *P. aeruginosa* cells in a liquid medium may be more interesting (since they are close to in vivo conditions) compared to growth on a solid medium. In this regard, the antimicrobial effects of peptides were tested on *P. aeruginosa* strains PA103 and ATCC 28753 (Figure 3 and Figure 4, respectively) by microdilution technique.

As seen in Figure 3, the peptides showed no significant antimicrobial activity compared to gentamicin sulfate after 24 h of incubation with the *P. aeruginosa* strain PA103 cells grown in a liquid medium. In order to take into account the antimicrobial effects dependent on a particular strain, we tested the same peptides R23R* and R23L* on various *P. aeruginosa* strains PA103 and ATCC 28753 (Figure 4).

The R23L* peptide is able to completely inhibits the growth of bacteria *P. aeruginosa* strain ATCC 28753 at concentrations of 6 µM, but not of the growth cells PA103 strain (Figure 4B).

### 2.3. Toxicity of R23R and R23L

Figure 5 shows the results of an in vitro toxicity test of the R23R and R23L peptides obtained to determine the viability of human fibroblasts (Figure 5A) and breast tumor cell lines (Figure 5B) after incubation with the peptides for 24 h.

To analyze the cytotoxic effect of peptides, cells after incubation with maximum peptide concentrations (8 µM) were stained with calcein-AM (stains living cells) and propidium iodide (stains dead cells). It was shown that all BT474 cells as well as fibroblasts remained alive after 24 h of incubation with the R23R and R23L peptides. The results of this work show that the studied peptides do not alter the viability of both BT474 human breast duct carcinoma cells and human fibroblasts at maximum concentrations (8 µM). The absence of the cytotoxic effect of the studied peptides was shown.

### 2.4. Amyloidogenic Properties of Synthesized Peptides

It is known that thioflavin T (ThT) has the ability to bind to amyloids and amyloid-like fibrils. This process is recorded by a multiple increase in the ThT fluorescence intensity at a wavelength of ~485 nm. ThT fluorescence spectra were obtained after 24 h of incubation at 37 °C of the dye with different concentrations of the R23R, R23L, R23R*, and R23L* peptides (Figure 6).

The effect of a multiple increase in the ThT fluorescence intensity after 24 h of incubation was obtained for the R23R (3600 µM), R23L (3800 and 380 µM), R23L* (3800, 380, and 38 µM) peptides. This effect was absent for the R23R* peptide at any tested concentration (≤3600 µM). Moreover, none of the peptides showed an increase in the relative intensity of ThT fluorescence at concentrations of 3.6–3.8 µM. In general, it can be concluded that the R23L and R23L* peptides had the greatest propensity to bind ThT.

## 3. Discussions

AMPs are usually characterized by a stable secondary structure, in particular, the presence of β-sheets and α-helices [68]. A decrease in the ordering of AMPs, in particular, caused by the replacement of amino acid residues with sarcosine, can lead to a decrease in the antimicrobial properties of modified peptides [64].

For the R23R, R23L, R23R*, and R23L* peptides, the secondary structure and physicochemical properties related to the potency and selectivity of AMPs were predicted (Table 1). According to the predictions of the JPred4 [65] and PredictProtein [66] programs, the synthesized R23R, R23L, R23R*, and R23L* peptides contain both disordered and ordered regions of the secondary structure. Thus, according to the JPred4 program, β-structure is predicted at the C-terminus of the R23R and R23L peptides. The replacement of glycine 12 with sarcosine leads to a contraction of the ordered region at the C-terminus of the R23R* peptide and but has no effect for the R23L* peptide.

Interestingly, according to the PredictProtein program, the replacement of glycine 12 with sarcosine in the central region of the R23R and R23L peptides leads to a contraction of the β-strand at the C-terminus of the modified R23R* and R23L* peptides. At the same time, the amino acid modification practically does not affect the results of helix prediction at the N-terminus of peptides, in the region corresponding to the CPP (Tat fragment). Thus, according to the predictions of the JPred4 and PredictProtein programs, the replacement of the glycine residue with sarcosine in the linker region of the peptide (“CPP”+”linker GGGG”+“predicted amyloidogenic-fragment”) affects the results of predicting the secondary structure of the peptide in its amyloidogenic region. In turn, a change in the secondary structure, can lead to a change in the antimicrobial action of peptides [69].

It is known that peptides with high hydrophobicity (H_n_) integrate better into the phospholipid bilayer of a bacterial cell, and a high hydrophobic moment (µH_n_) allows peptides to form a water channel, which, in turn, can enhance their antimicrobial effect. H_n_ and µH_n_ are closely related to the general amphiphilicity, the ability of peptides to aggregate in solution, and also to interact with cell membranes [70,71]. However, an excessively high value of H_n_ is already associated with disordered aggregation of peptides, as well as with an increase in their toxicity towards eukaryotic cells [72,73]. It was shown that a decrease in H_n_ and µH_n_ increases antimicrobial selectivity, but also decreases the overall antimicrobial activity of peptides [74]. It can be expected that the modified R23R* and R23L* peptides, in comparison with R23R and R23L, will exhibit a lower antimicrobial effect, but a higher selectivity with respect to different phospholipid composition of different bacteria (species and strains). The charge of peptide (Z) is important both for interaction with the bacterial membrane and for loading the peptide into the carrier [75]. For interaction with a negatively charged bacterial membrane, it is preferable that the antimicrobial peptide has a sufficiently high positive charge (Z = +5–+12). The isoelectric point (I_0_) of the peptide should have a value that is very different from the pH value of the buffer solution (I_0_ > 9), in which the antimicrobial properties of the peptides are tested (as a rule, the pH value is 7–8, close to the physiological conditions of a bacterial cell). As shown in Table 1, modification glycine 12 does not appear to affect the Z and I_0_ values for the R23R, R23L, R23R*, and R23L* peptides. At the same time, the peptides have Z (+9 for R23R and +7 for R23L) and I_0_ (12.51 for R23R and 12.41 for R23L), which contribute to the manifestation of antimicrobial effects [32].

Data on the secondary structure and physicochemical properties of peptides, in turn, can be used to predict their antimicrobial properties using existing bioinformatics tools [76,77]. In Table 2 the antimicrobial activity was assessed using four algorithms from the CAMPR3 online server: random forests (RF), support vector machine (SVM), artificial neural network (ANN), and discriminant analysis (DA). The prediction results are highly dependent on the algorithm used. Thus, DA predicted antimicrobial properties for all four peptides R23R, R23L, R23R*, and R23L*. At the same time, the SVM algorithm did not find any antimicrobial peptides among the synthesized peptides. The AmpGram tool predicted R23R and R23R* would be more suitable as antimicrobial peptides than R23L and R23L*. At the same time, AMP Scanner favored R23L and R23L* as antimicrobials over R23R and R23R*, which had no predictable antimicrobial properties. According to the predictions of the online servers CAMPR3, AmpGram, AMP Scanner, replacing glycine 12 with sarcosine did not significantly change the predictions of antimicrobial properties of the modified peptides.

Testing of the peptides on *P. aeruginosa* strains revealed antimicrobial activity that is specific for the strain and growth condition. R23R and R23L were active against both strains PA103, and ATCC 28753 with the MIC of about 300 µM, when tested on agar medium (Figure 2 and Table 3 respectively). For comparison, the MIC of the antibiotic gentamicin sulfate was 170 µM. The modified peptides R23R* and R23L* had no bactericidal effect against bacteria grown on agar. *P. aeruginosa* strain ATCC 9027 used in the study was the least sensitive to the action of the synthesized peptides.

*P. aeruginosa* strains behaved differently when growing in a liquid medium (Figure 3 and Figure 4). PA103 was generally resistant to all peptides. In contrast, the R23L* peptide inhibited the growth of ATCC 28753 cells at low concentrations of 6 and 12 µM. These effects, depending on the strain and cultivation condition, can be due to strain-specific differences and the well-known morphologic plasticity of *P. aeruginosa*, which provides the appearance of morphotypes that differ in sensitivity to antibiotics, the presence or absence of exoenzymes, pigmentation, and antigenicity even within the same culture [78]. Experiments on spontaneous aggregation demonstrated that the ability of cells to form unbound aggregates strongly depends on the bacterial strain [79].

At the same time the antibacterial properties of the R23L and R23L* peptides are higher than those of R23R and R23R*. This effect can be associated with different cytotoxicity of peptides towards *P. aeruginosa* cells based on the different amyloidogenicity of the sequence of the R23R, R23L, R23R*, and R23L* peptides. All peptides were developed for targeted coaggregation with the S1 protein from *P. aeruginosa*; therefore, their antimicrobial (cytotoxic) action should be specific for these bacterial cells and not on eukaryotic cells. It is known that antimicrobial peptides can exhibit differential cytotoxicity to various types of mammalian cell lines [80]. We tested the differences in the cytotoxicity of peptides R23R, R23R*, R23L, and R23L* in relation to human fibroblast and carcinoma cells (Figure 5). According to the results of such tests, it was revealed that the peptides did not have a toxic effect on eukaryotic cells at the tested concentrations (≤8 μM).

As discussed in the Introduction, the R23L and R23L* peptides were developed based on the amyloidogenic sequence ITDFGIFIGL, which forms amyloid-like fibrils, while R23R and R23R* did contain amyloidogenic sequence, however they failed to identify amyloidogenic properties through their conducted studies [51]. Thus, as can be seen from the results of measuring ThT fluorescence (Figure 6), the R23L and R23L* peptides are more prone to amyloidogenesis than R23R and R23R*. In an earlier study [26], in which the amyloidogenic and antimicrobial effects of peptides based of the S1 protein from the model organism *T. thermophilus* were assessed from two hybrid peptides (R23I and R23T), R23I showed a higher antimicrobial effect. Interestingly, R23I in comparison with R23T, as well as R23L in comparison with R23R, was more amyloidogenic. Increasing the length of peptides and proteins can reduce their amyloidogenicity (the number of amino acid residues predicted to be amyloidogenic), as was discussed in [81]. At the same time, it was shown that whole proteins with a molecular weight of more than 60 kDa can exhibit an antimicrobial effect [82]. We have previously shown that peptide modification decreases the ability of peptides to form fibrils, but increases the ability of peptides to suppress the growth of *T. thermophilus* cell culture [26]. Previously, similar results were reported, showing that the polymorphism of fibrils of amyloidogenic peptides [83] can have a strong influence on its antimicrobial effects and variable cytotoxicity [84].

It is noteworthy that the synthesized peptides did not show the same effectiveness against various strains of *P. aeruginosa*. In this regard, it is necessary to develop new modified hybrid AMPs in order to improve their antimicrobial properties, in particular, to increase the half-life in the bacterial cell, potency in coping with persister cells, as well as to prevent biofilm formation. Therefore, of great interest is the work [85], in which the properties of cathelicidin-BF15 were significantly improved by changing the composition of the original amino acid sequence and cyclization of the peptide. The modified ZY4 peptide proved to be an ideal candidate not only against different strains of *P. aeruginosa,* but also against different species of pathogens. Of course, it is necessary to compare the antimicrobial effect of different peptides on the same strain and under the same conditions. In our case, the R23L* peptide showed a significant antimicrobial effect against ATCC 28753 strain at a concentration of 6–12 μM comparable with gentamicin sulfate (Figure 4) and did not show a significant antimicrobial effect against the PA103 strain at a concentration of 380 μM (Figure 3). Moreover, for the known antimicrobial peptide Mel4 [86], differences in effective concentrations were shown when tested on various strains of *P. aeruginosa*. However, the differences in MIC are not as large as for our R23L* peptide. At the same time, it should be noted that Mel4 is very active against *P. aeruginosa* due to the destruction of cell membranes and lysis of bacteria [87]. In turn, the main mechanism of action for R23L* is assumed to be different, associated with directed coaggregation with the ribosomal S1 protein. R23L* is interesting from the point of view of the use of antimicrobial peptides with different mechanisms of action. It can be expected that bacteria will develop resistance to the R23L* peptide for a longer time. In the future, it is planned to determine the effect of amino acid modification (G12-> Sar12) in the non-amyloidogenic region of peptides on the change in the proteomic composition of various *P. aeruginosa* strains treated with the R23R, R23L, R23R*, and R23L* peptides. At the same time, the antimicrobial effect of peptides R23L and R23L* is comparable with the commercial antibiotic gentamicin sulfate. The antimicrobial effect can be enhanced by increasing the overall amyloidogenicity of new peptides synthesized based on the S1 protein sequence from *P. aeruginosa* and by considering other cell penetrating peptides.

## 4. Materials and Methods

### 4.1. Synthesis and Characterization of Peptides

#### 4.1.1. Peptide Synthesis

Peptides RKKRRQRRRGGGGLHITDMAWKR (R23R, 2820.3 Da) and RKKRRQRRRGGGGITDFGIFIGL (R23L, 2645.1 Da) were commercial products (IQ Chemical LLC, S. Petersburg, Russia). Peptides RKKRRQRRRGG-Sar-GLHITD-Nle-AWKR (R23R*, 2815.3 Da), RKKRRQRRRGG-Sar-GITDFGIFIGL (R23L*, 2658.1 Da), and RKKRRQRRRGG-Sar (R12-Sar, 1523.8 Da) were synthesized in-house using routine techniques of Fmoc/^t^Bu SPPS methodology [61]. Semipreparative RP-HPLC purification was carried out in isocratic mode (mobil phase “A” 0.1% TFA in water, mobil phase “B” acetonitrile (no additives), 22% “B” for R23R* and 32% B for R23L*) on Luna C18 250 × 21.5 mm (10 µm) column (Phenomenex, Torrance, CA, USA) using Gilson instrument equipped with model 305/302 binary pump, model 803 manometric module and model 811 dynamic mixer at flow rate 10 mL/min (Gilson, Middleton, WI, USA). The fractions collected were analyzed using RP-HPLC on Luna 5 µm C18 (2) 100 Å 250 × 4.6 column (Phenomenex, Torrance, CA, USA) using Waters instrument equipped with Waters 2487 Dual Absorbance Detector and Waters 1525 Binary HPLC Pump (Waters, Milford, MA, USA). The appropriate fractions were lyophilized and the peptide identity was confirmed using an Orbitrap Elite mass spectrometer (Thermo Scientific, Dreieich, Germany). The estimated peptide molecular weight coincided with the calculated one.

#### 4.1.2. Bioinformatic Analysis of Peptides

Amyloidogenic regions of protein bPaS1 from *P. aeruginosa* with a length of at least five amino acid residues were determined using four programs—AGGRESCAN, FoldAmyloid, Pasta 2.0, and Waltz [51]. Predicted amyloidogenic regions were checked in vitro by their ability to form amyloid-like aggregates and two sequences (LHITDMAWKR, ITDFGIFIGL) were used as “amyloidogenic” parts for the R23R, R23L, R23R*, and R23L* peptides.

Default software settings by JPred4 [65], PredictProtein [66], DBAASP v3.0 [67], CAMPR3 [9], AmpGram [7], and AMP Scanner [8] were used for the prediction of secondary structure, physicochemical and antimicrobial properties of R23R, R23L, R23R*, and R23L*.

The JPRED4 program makes secondary structure prediction of protein sequences by the JNET algorithm (jnetpred), http://www.jalview.org/help/html/webServices/jnet.html (accessed on 2 August 2021) [65]. PredictProtein provides secondary structure prediction (helix, strand, and other) for each amino-acid residue in the query by according the RePROF tool, https://predictprotein.org/ (accessed on 2 August 2021) [66,88]. Among the elements of the secondary structure predicted by the JPred4 and PredictProtein programs, fragments of peptides with a length of at least 5 amino acid residues were selected (just as the amyloidogenic regions were previously selected by length).

The “Moon and Fleming” (MF) hydrophobicity scale of the “Property Calculation” tool of the Database of Antimicrobial Activity and Structure of Peptides (DBAASP v3.0) was used to predict the physicochemical characteristics of peptides, https://dbaasp.org/property-calculation (accessed on 2 August 2021) [67,89]. All algorithms of the “Predict Antimicrobial Peptides” tool of Collection of Anti-Microbial Peptides (database CAMPR3) were used to predict AMP probability for the R23R, R23L, R23R*, and R23L* peptides, http://www.camp.bicnirrh.res.in/predict/ (accessed on 2 August 2021) [9]. The effect of replacement one residue in the peptides (glycine 12 with proline or alanine residues as similar to sarcosine in structure and properties) was studied using four algorithms, namely random forest (RF), support vector machine (SVM), artificial neural network (ANN), and discriminant analysis (DA) [90].

AmpGram uses the RF algorithm based on amino acid motifs (n-grams) encoded peptides for calculation of AMP probability, http://biongram.biotech.uni.wroc.pl/AmpGram/ (accessed on 2 August 2021) [7]. AMP Scanner vr.2 proposes a deep neural network (DNN) classifier for predicting AMP from sequence alone a peptide/protein, https://www.dveltri.com/ascan/v2/ascan.html (accessed on 2 August 2021) [8].

A peptide is classified as AMP if the probability is equal to or greater than 0.5, and the peptide is classified as non-AMP if the probability is less than 0.5 based on prediction results using the CAMPR3, AmpGram, AMP Scanner tools.

### 4.2. Antimicrobial Activity of Peptides

#### 4.2.1. Determination of the Antibacterial Properties of Peptides on Agar

The R23R, R23L, R23R*, R23L* peptides and an antibiotic (gentamicin sulfate) were dissolved in 100% DMSO and solutions were prepared to test their activity.

The culture of the studied bacteria (*P. aeruginosa* strain PA103) was diluted 10 times in Luria-Bertani (LB) medium with non-solidified 0.75% agar (w/w) on sterile Petri dishes. Drops of the sample (a solution of peptide or gentamicin sulfate with a final concentration of DMSO) 20% (volume/volume) at concentrations: 1—peptide 3600–3800 µM (gentamicin sulfate 17,000 µM), 2—peptide 360–380 µM (gentamicin sulfate 1700 µM), 3—peptide 36–38 µM (gentamicin sulfate 170 µM), 4—peptide 3.6-3.8 µM (gentamicin sulfate 17 µM), are applied to the surface of agar plates, and after 24 h incubation, the result is assessed by the presence of zones of inhibition.

The *P. aeruginosa* strain ATCC 28753 was inoculated with a wire loop in 5 mL of LB medium and was incubated at 37 °C. After 12 h, plastic Petri dishes were poured with LB medium (about 5 mL) with 0.75% agar. Before pouring, 50 μL of an overnight culture of the test bacteria was added to each dish. The R23R, R23L, R23R*, and R23L* peptides were finally dissolved in 2% (volume/volume) DMSO and was tested with different concentrations with gentamicin sulfate as control of antimicrobial effect. The medium was dried and 10 μL of the prepared peptides were applied to it in accordance with the labeling. Then the dishes were placed in a thermostat and incubated for 12 h at a temperature of 37 °C. Antimicrobial activity was recorded by the presence of transparent zones of no growth around drops with peptide.

Studies with the *P. aeruginosa* strain ATCC 9027 were carried out similarly to the *P. aeruginosa* strain ATCC 28753. The exception was the amount of LB medium. Sterile plastic Petri dishes were filled with LB medium (about 25 mL) with 0.75% agar and 250 µL of an overnight culture of the test bacteria was added to each dish.

#### 4.2.2. Determination of MIC by Broth Dilution Method

The R23R, R23L, R23R*, R23L* peptides and an antibiotic (gentamicin sulfate) were dissolved in 100% DMSO and solutions were prepared with a final DMSO concentration of 2% (volume/volume) to test their activity.

Stationary culture of *P. aeruginosa* strain PA103 was diluted 100 times in fresh LB medium with peptides R23R and R23L or gentamicin sulfate (positive control) at various concentrations. After 24 h at 37 °C co-incubation bacterial cells with peptides, the optical densities of the cultures were measured at a wavelength of 600 nm in a plate reader iMac (BioRad Laboratories, Hercules, CA, USA). Each test was carryout in three replicates.

In the next experiment to prepare a stationary culture of *P. aeruginosa*, strain PA103, a colony was inoculated from a Petri dish in 1.5 mL of liquid LB medium and set to grow for 24 h at 37 °C and with stirring at 180 rpm. In the experiment, we used a culture with an initial concentration of ≈10^6^–10^7^ CFU/mL (10 μL), sterile LB medium (80 μL), peptides R23R* and R23L* diluted in 20% (volume/volume) DMSO or the antibiotic gentamicin sulfate as a positive control (10 μL). Final concentrations of peptides 12 μM, 6 μM, 3 μM, 1.5 μM, and 0.75 μM and gentamicin sulfate are 55.6 μM, 27.8 μM, 13.9 μM, 7 μM, 3.5 μM. As a control, we used: (1) a culture diluted 1000 times with a medium without peptides and antibiotics with the addition of sterile physical. Solution (10 μL); (2) culture diluted 1000 times liquid LB medium with DMSO at a final concentration of 2% (10 μL). The samples were grown for 10 h at 37 °C in the wells of flat-bottom sterile 96-well Linbro plastic plates (Flow Laboratories; McLean, VA, USA). Optical density was measured in a plate reader iMac (BioRad Laboratories, Hercules, CA, USA) at a wavelength of 600 nm every hour from 0 h to 24 h. Each sample is measured in duplicate.

Determination of the MIC for *P. aeruginosa* strain ATCC 28753 was performed using Mueller-Hinton Broth (MHB) (Sigma-Aldrich, St. Louis, MO, USA). The composition of the reaction medium: 80 μL MHB; 10 μL bacteria at a concentration of ≈ 10^6^–10^7^ CFU/mL; peptide (each peptide was dissolved to a final concentration of 12 μM, 6 μM, 3 μM, 1.5 μM, and 0.75 μM, respectively). *P. aeruginosa* cells were incubated with or without a peptide in a 96-well microtiter plate for 16–24 h, data was taken every 30 min. Bacterial growth was assessed by spectrophotometry using a spectrophotometer Multiskan GO (Thermo Scientific, Waltham, MA, USA). The effect of gentamicin sulfate on the cell culture growth served as a positive control. Sterile MHB with 2% (volume/volume) final concentration of DMSO and the tested bacteria strain without the addition of peptide were used as negative controls.

### 4.3. Cytotoxicity Assay

Cell viability was estimated as described previously by resazurin cell viability assay [26,91]. Each assay was done in triplicate. All measurements were carried out on the control samples that were not treated with cytotoxic substances. We used human diploid fibroblasts and human breast duct carcinoma BT474 cells obtained from the American Type Cell Culture Collection (Manassas, Virginia, VA, USA). BT474 cells were cultured in DMEM medium (Sigma-Aldrich, St. Louis, MO, USA) supplemented with 10% fetal bovine serum (Gibco, Waltham, MA, USA), 40 μg/mL gentamicin sulfate (Sigma-Aldrich, St. Louis, MO, USA), at 37° C, under conditions of 5% content CO_2_ in the air. The analysis of living and dead cells was carried out using BD Accuri C6 flow cytometer (BD Bioscience, San Jose, CA, USA).

### 4.4. Thioflavin T Fluorescence Measurement

Preparations of the R23R, R23L, R23R*, and R23L* peptides in buffer conditions 50 mM TrisHCl, pH 7.5; 150 mM NaCl, 20% (volume/volume) DMSO was incubated with 200 µM thioflavin T (ThT) at 37 °C with shaking for 24 h at 450 rpm in a thermostatic mixer Thermomixer comfort (Eppendorf, Hamburg, Germany). The spectra of fluorescence intensity of free ThT and in solution with individual peptides were studied by us using the method of fluorescence spectroscopy as described previously [51].

### 4.5. Statistical Analysis

The obtained results were statistically manipulated with OriginPro (OriginLab Corporation, Northampton, MA, USA) software. Data are presented as the mean ± standard deviation (M ± SD). The experiments were performed in at least two repetitions (*n* ≥ 2). The statistical significance of the difference was determined using the Student’s t-test and analysis of variance (ANOVA).

## 5. Conclusions

In this study, we have developed and synthesized antimicrobial peptides R23R, R23L, as well as their analogs R23R*, R23L*. The secondary structure, antimicrobial and amyloidogenic properties of the peptides were predicted and evaluated. The combination of the bioinformatic, microbiological, and physical methods used made it possible to assess the antimicrobial properties of the R23R, R23L, R23R*, and R23L* peptides, as well as the effect on these properties of only one amino acid modification (glycine at 12 position is replaced with sarcosine). The R23L and R23L* peptides have higher antimicrobial activity than the R23R and R23R* peptides, and the MIC for R23L* was comparable to the antibiotic gentamicin sulfate for the P. aeruginosa strain ATCC 28753.

## Figures and Tables

**Figure 1 ijms-22-09776-f001:**
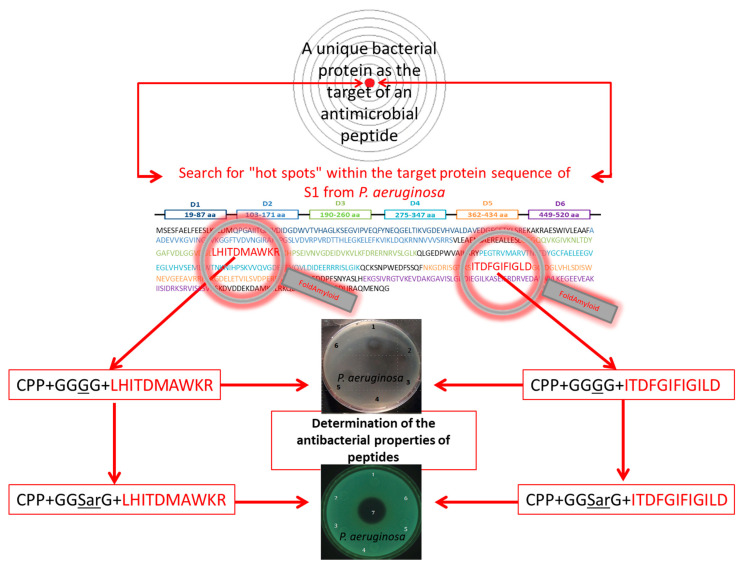
Schematic diagram of the creation of new AMPs (against *P. aeruginosa*) based on targeted protein aggregation. The domain organization of the S1 protein is shown schematically, each of the six protein domains (D1–D6) is color represented. The red font shows two amyloidogenic regions: LHITDMAWKR from the third domain and ITDFGIFIGLD from the fifth domain predicted by the FoldAmyloid and AGGRESCAN programs within the ribosomal S1 protein from *P. aeruginosa* [51]. The N-terminus of synthesized hybrid peptides contains a cell penetrating peptide (CPP) fragment RKKRRQRRR connected to the amyloidogenic region using a linker of four glycine residues (GGGG) or three glycine and sarcosine residues (GG-Sar-G). The hybrid AMPs with different concentrations can be tested against *P. aeruginosa* strains (see below “Materials and Methods” section).

**Figure 2 ijms-22-09776-f002:**
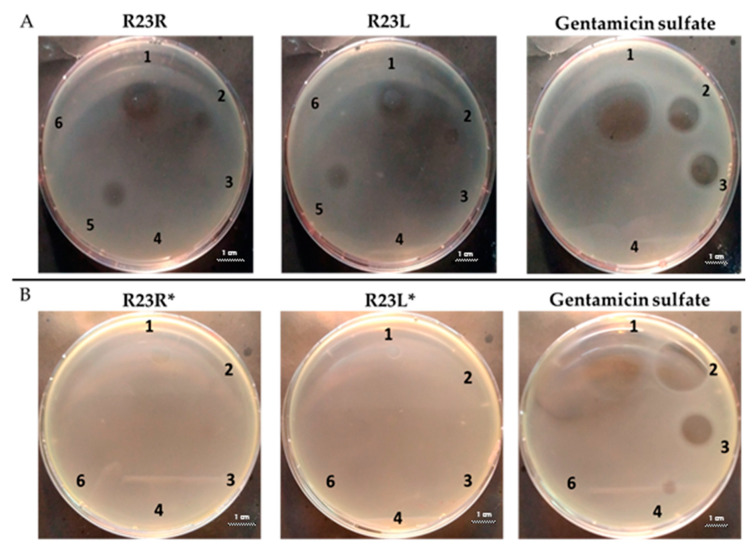
Inhibition zone assay after 24 h of co-incubation peptides R23R, R23L (**A**) and modified peptides R23R*, R23L* (**B**) with the cells *P. aeruginosa* (strain PA103). The test for the antibacterial effect of peptides was carried out according to the following scheme: **1**—peptide 3600–3800 µM (gentamicin sulfate 17,000 µM), **2**—peptide 360–380 µM (gentamicin sulfate 1700 µM), **3**—peptide 36–38 µM (gentamicin 170 µM), **4**—peptide 3.6–3.8 µM (gentamicin sulfate 17 µM), **5**—DMSO 100% (volume/volume), **6**—DMSO 20% (volume/volume). The scale bar is 1 cm.

**Figure 3 ijms-22-09776-f003:**
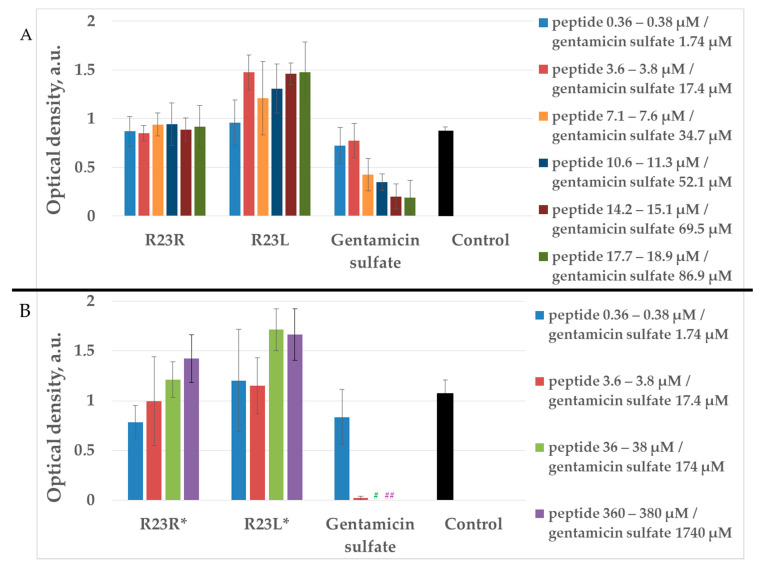
Antibacterial activity of R23R, R23L (**A**) and modified peptides R23R*, R23L* (**B**) after 24 h of co-incubation the peptides with the *P. aeruginosa* (strain PA103) cells. Various concentrations of gentamicin sulfate were used as positive control for the antimicrobial effect. Control is LB medium containing cell culture in 2% (volume/volume) DMSO without peptide and gentamicin sulfate. Error bars show standard errors. ^#^ Optical density is equal to zero for 174 µM concentration of gentamicin sulfate. ^##^ Optical density is equal to zero for 1740 µM concentration of gentamicin sulfate.

**Figure 4 ijms-22-09776-f004:**
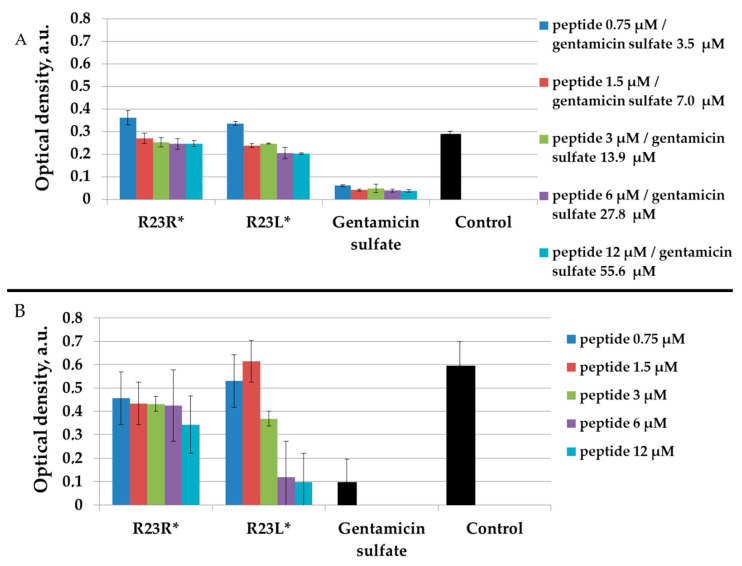
Inhibitory activity of R23R* and R23L* during 16 h of co-incubation the peptides with the cells *P. aeruginosa* PA103 (**A**) and ATCC 28753 (**B**). Different concentrations (**A**) and only 1700 µM (**B**) of gentamicin sulfate were used as positive control of antimicrobial effect. The tested bacteria strain without the addition of peptide were used as negative control.

**Figure 5 ijms-22-09776-f005:**
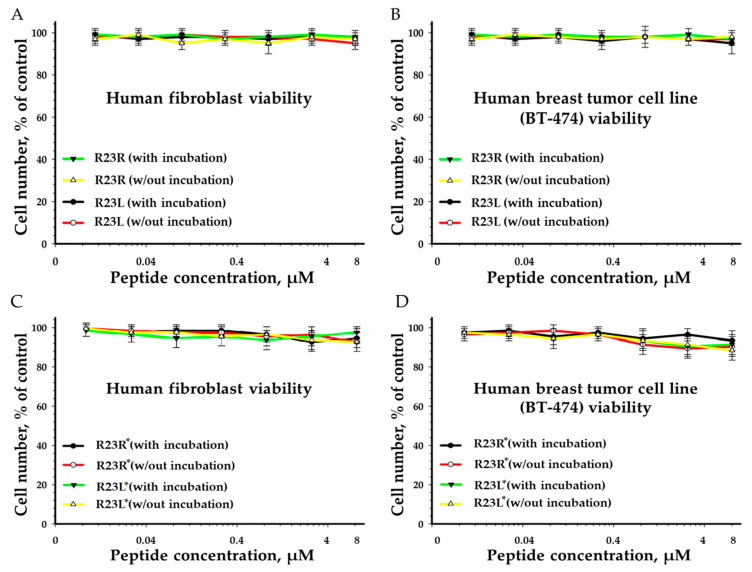
Effects of peptide treatment on survival the human fibroblasts (**A**,**C**) and breast tumor cell line BT-474 (**B**,**D**). Error bars show standard errors.

**Figure 6 ijms-22-09776-f006:**
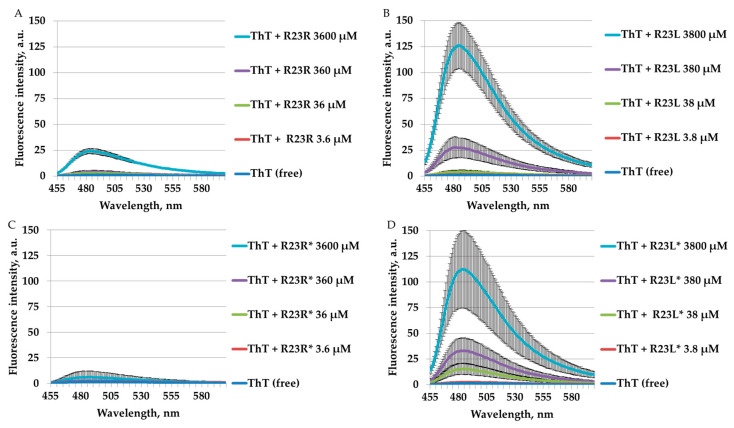
Spectra of fluorescence intensity of free thioflavin T (ThT) and in solution with individual peptides R23R (**A**), R23L (**B**), R23R* (**C**), and R23L* (**D**) under conditions of 50 mM TrisHCl, 150 mM NaCl, pH 7.5, 20% (volume/volume) DMSO after 24 h at 37 °C. Error bars with standard deviations for the mean values of the measured fluorescence intensity are shown.

**Table 1 ijms-22-09776-t001:** Prediction of the secondary structure ^§^ and physicochemical properties of the peptides.

Peptide	JPred4 (Jnetpred)	Predict Protein (RePROF)	DBAASP v3.0 (MF Scale)			
			µH_n_	H_n_	Z	I_0_
R23R vs. R23R*						
RKKRRQRRRGGGGLHITDMAWKR	β-strand 14–18 a.a. (LHITD)	Helix 2–9 a.a. (KKRRQRRR), β-strand 14–18 a.a. (LHITD)	0.36	0.91	+9	12.51
RKKRRQRRRGG-Sar ^§§^(A)-GLHITD-Nle ^§§§^(M)-AWKR		Helix 2–9 a.a. (KKRRQRRR), other 10–14 a.a. (GGAGL), helix 18–22 (DMAWK)	0.32	0.84	+9	12.51
RKKRRQRRRGG-Sar(A)-GLHITD-Nle(L)-AWKR	β-strand 14–18 a.a. (LHITD)	Helix 2–9 a.a. (KKRRQRRR), other 10–14 a.a. (GGAGL), helix 18–22 (DMAWK)	0.29	0.79	+9	12.51
RKKRRQRRRGG-Sar(P)-GLHITD-Nle(M)-AWKR		Helix 2–9 a.a. (KKRRQRRR), other 10–14 a.a. (GGAGL)	0.29	0.77	+9	12.51
RKKRRQRRRGG-Sar(P)-GLHITD-Nle(L)-AWKR		Helix 2–9 a.a. (KKRRQRRR), other 10–14 a.a. (GGPGL)	0.27	0.73	+9	12.51
R23L vs. R23L*						
RKKRRQRRRGGGGITDFGIFIGL	β-strand 17–21 a.a. (FGIFI)	Helix 3–9 a.a. (KRRQRRR), other 10–15 a.a. (GGGGIT), β-strand 16–21 a.a. (DFGIFI)	0.24	0.18	+7	12.41
RKKRRQRRRGG-Sar(A)-GITDFGIFIGL	β-strand 17–21 a.a. (FGIFI)	Helix 3–9 a.a. (KKRRQRRR), other 10–15 a.a. (GGAGIT)	0.18	0.11	+7	12.41
RKKRRQRRRGG-Sar(P)-GITDFGIFIGL	β-strand 17–21 a.a. (FGIFI)	Helix 3–9 a.a. (KKRRQRRR), other 10–15 a.a. (GGPGIT)	0.14	0.04	+7	12.41

^§^ The predicted secondary structure elements of length ≥5 a.a. ^§§^ For calculations, sarcosine (Sar) was replaced by analogs similar in properties and structure: alanine (A) and proline (P). ^§§§^ For calculations, norlecine (Nle) was replaced by analogs of similar properties and structure: methionine (M) and leucine (L). µH_n_ is the normalized hydrophobic moment, H_n_ is the normalized hydrophobicity, Z is the value of charge at pH 7, I_0_ is the value of isoelectric point.

**Table 2 ijms-22-09776-t002:** Antimicrobial prediction for the R23R, R23L, R23L*, and R23L* peptides.

Peptide	CAMPR3				AmpGram	AMP Scanner
	RF	SVM	ANN	DA	RF and n-Grams	DNN
R23R vs. R23R*						
RKKRRQRRRGGGGLHITDMAWKR	0.48 (non-AMP ^§^)	0.03 (non-AMP)	AMP ^§§^	0.93 (AMP)	0.59 (AMP)	0.07 (non-AMP)
RKKRRQRRRGG-Sar ^§§§^(A)-GLHITD-Nle ^§§§§^(M)-AWKR	0.49 (non-AMP)	0.03 (non-AMP)	AMP	0.95 (AMP)	0.54 (AMP)	0.05 (non-AMP)
RKKRRQRRRGG-Sar(A)-GLHITD-Nle(L)-AWKR	0.58 (AMP)	0.05 (non-AMP)	AMP	0.96 (AMP)	0.63 (AMP)	0.03 (non-AMP)
RKKRRQRRRGG-Sar(P)-GLHITD-Nle(M)-AWKR	0.48 (non-AMP)	0.04 (non-AMP)	non-AMP	0.95 (AMP)	0.47 (non-AMP)	0.13 (non-AMP)
RKKRRQRRRGG-Sar(P)-GLHITD-Nle(L)-AWKR	0.53 (AMP)	0.07 (non-AMP)	AMP	0.97 (AMP)	0.62 (AMP)	0.07 (non-AMP)
R23L vs. R23L*						
RKKRRQRRRGGGGITDFGIFIGL	0.59 (AMP)	0.06 (non-AMP)	AMP	1.00 (AMP)	0.37 (non-AMP)	0.93 (AMP)
RKKRRQRRRGG-Sar(A)-GITDFGIFIGL	0.59 (AMP)	0.06 (non-AMP)	AMP	1.00 (AMP)	0.22 (non-AMP)	0.44 (non-AMP)
RKKRRQRRRGG-Sar(P)-GITDFGIFIGL	0.58 (AMP)	0.10 (non-AMP)	AMP	1.00 (AMP)	0.45 (non-AMP)	0.95 (AMP)

^§^ It is predicted as a peptide that does not exhibit antimicrobial activity. The prediction level is less than 0.5. ^§§^ It is predicted as a peptide with antimicrobial activity. The prediction level is over than 0.5. ^§§§^ For calculations, sarcosine (Sar) was replaced by analogs similar in properties and structure: alanine (A) and proline (P). ^§§§§^ For calculations, norlecine (Nle) was replaced by analogs of similar properties and structure: methionine (M) and leucine (L). Antimicrobial activity was predicted using algorithms: RF is a random forest, SVM is a support vector machine, ANN is an artificial neural network, DA is a discriminant analysis, RF and n-gramms is a random forest based on amino acid motifs, DNN is deep neural network.

**Table 3 ijms-22-09776-t003:** Results of testing the antimicrobial properties of the R23R, R23L, R23R*, and R23L* peptides with the *P. aeruginosa* (strain ATCC 28753) cells. The scale bar is 1 cm.

Photo of Results	Scheme
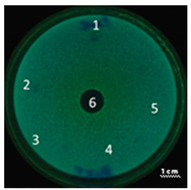	Test of peptide R23R 1—R23R, 300 µM 2—R23R, 250 µM 3—R23R, 200 µM 4—R23R, 150 µM 5—DMSO, 2% (volume/volume) 6—Gentamicin sulfate, 1700 µM
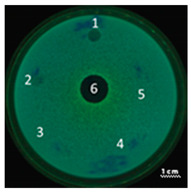	Test of peptide R23L 1—R23L, 300 µM 2—R23L, 150 µM 3—R23L, 75 µM 4—R23L, 37.5 µM 5—DMSO, 2% (volume/volume) 6—Gentamicin sulfate, 1700 µM
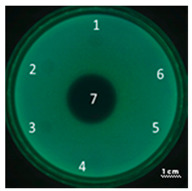	Test of peptide R23R* 1—R23R*, 300 µM 2—R23R*, 150 µM 3—R23R*, 75 µM 4—R23R*, 37.5 µM 5—R23R*, 18.8 µM 6—DMSO, 2% (volume/volume) 7—Gentamicin sulfate, 1700 µM
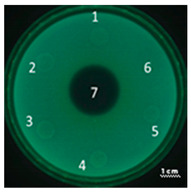	Test of peptide R23L* 1—R23L*, 300 µM 2—R23L*, 150 µM 3—R23L*, 75 µM 4—R23L*, 37.5 µM 5—R23L*, 18.8 µM 6—DMSO, 2% (volume/volume) 7—Gentamicin sulfate, 1700 µM

**Table 4 ijms-22-09776-t004:** Results of testing the antimicrobial properties of the R23R, R23L, R23R*, and R23L* peptides with the *P. aeruginosa* (strain ATCC 9027) cells. The scale bar is 1 cm.

Photo of Results	Scheme
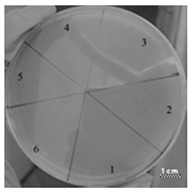	Test of peptide R23R 1—R23R, 3750 µM 2—R23R, 375 µM 3—R23R, 37.5 µM 4—R23R, 3.75 µM 5—DMSO, 20% (volume/volume) 6—LB medium
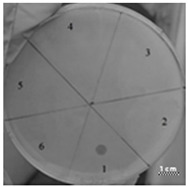	Test of peptide R23L 1—R23L, 3750 µM 2—R23L, 375 µM 3—R23L, 37.5 µM 4—R23L, 3.75 µM 5—DMSO, 20% (volume/volume) 6—Luria-Bertani (LB) medium
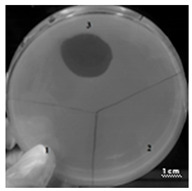	1—LB medium 2—DMSO, 20% (volume/volume) 3—Gentamicin sulfate, 17 µM
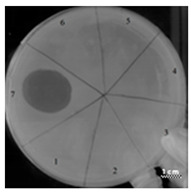	Test of peptide R23R* 1—R23R*, 3750 µM 2—R23R*, 375 µM 3—R23R*, 37.5 µM 4—R23R*, 3.75 µM 5—DMSO, 20% (volume/volume) 6—LB medium 7—Gentamicin sulfate, 17 µM
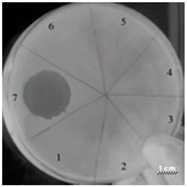	Test of peptide R23L* 1—R23L*, 3750 µM 2—R23L*, 375 µM 3—R23L*, 37.5 µM 4—R23L*, 3.75 µM 5—DMSO, 20% (volume/volume) 6—LB medium 7—Gentamicin sulfate, 17 µM

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
