# Peer review of "Is It Possible to Create Antimicrobial Peptides Based on the Amyloidogenic Sequence of Ribosomal S1 Protein of P. aeruginosa?"

_ijms, 2021, doi:10.3390/ijms22189776_

Round 1

Reviewer 1 Report

The manuscript entitled " Is it possible to create antimicrobial peptides based on the amyloidogenic sequence of ribosomal S1 protein of P. aeruginosa?" by Grishin et al describe the use of novel hybrid amyloidogenic peptide as novel antimicrobial therapeutics against P. aeruginosa. The strength  of the study is to design a hybrid peptide based on the amyloidogenic region of S1 ribosomal protein and integrating it with a cell penetrating peptide thus increasing the overall permeability of the peptide and also its antimicrobial properties. However, there are several weakness needs to be addressed before it is ready for publication as discussed below:

1) The whole manuscript is poorly written with lots of grammatical error which makes it hard to understand for the reader. There are many examples butt not least like Line 67-68, Line 78-90, Line 98-99 in Introduction section which needs to be simplify. Similarly, while discussing the result section the authors needs to explain the objective of each experiment performed and the expected outcome which they failed to do so.

2) Figure 1 : It would be better to represent the S1 protein also as a schematic representation so that the readers can better understand its domain organization. Also please elaborate the Figure 1 legend.

3)  Prediction of the Secondary Structure and Antimicrobial Propensities of the R23R, R23L Although the authors tried to perform some bioinformatic analysis to inform the secondary structure of their synthesized peptides, I think it would be nice to add some CD experiments to prove the same. Computational analysis can be some what misleading and addition of CD experiments can be finally complemented with these algorithmic analysis.

4) Also please explain the results clearly as observed from Table 2.

5)Experimental Cheking of the Antibacterial Activity of Peptides : For the agar plate experiments no negative control was used. I think the authors can simply use a scrambled peptide sequence or any peptide sequence which does not have any amyloidogenic properties as negative control. Also, they failed to compare the AMP effect of their synthesized peptides with already tested AMP against P. aeruginosa, for e.g. peptide ZY4 (Mwangi et al, PNAS, the test AMP in this study would have fared against the already known AMPs.

6) Measurement of the Antibacterial Activity of Peptides by Microdilution Technique: Figure 3:   Optical density decreases with decrease in the AMP concentration. Does this means test AMPs are more potent at lower concentration?

7) Toxicity of R23R and R23L: Why doesn't the authors included R23R and R23L in toxicity study especially when they have established that R23L* have more potent AMP activity as compared to others. Also, what is the premise to choose 8uM concentration for toxicity study.

8) Discussion: Please refer to any table or figures in the result section when you discuss about the results. Also, please explain the observation more clearly and state the advantages of these current AMPs over the already studied and published AMPs.

Author Response

The manuscript entitled " Is it possible to create antimicrobial peptides based on the amyloidogenic sequence of ribosomal S1 protein of P. aeruginosa?" by Grishin et al describe the use of novel hybrid amyloidogenic peptide as novel antimicrobial therapeutics against P. aeruginosa. The strength  of the study is to design a hybrid peptide based on the amyloidogenic region of S1 ribosomal protein and integrating it with a cell penetrating peptide thus increasing the overall permeability of the peptide and also its antimicrobial properties. However, there are several weakness needs to be addressed before it is ready for publication as discussed below:

1) The whole manuscript is poorly written with lots of grammatical error which makes it hard to understand for the reader. There are many examples butt not least like Line 67-68, Line 78-90, Line 98-99 in Introduction section which needs to be simplify. Similarly, while discussing the result section the authors needs to explain the objective of each experiment performed and the expected outcome which they failed to do so.

Response: Thank you for the suggested corrections and helpful comments to our manuscript. All suggestions and wishes were taken into account by us and included in the revised version of the manuscript.

2) Figure 1: It would be better to represent the S1 protein also as a schematic representation so that the readers can better understand its domain organization. Also please elaborate the Figure 1 legend.

Response: Figure 1 has been revised as recommended.

3)  Prediction of the Secondary Structure and Antimicrobial Propensities of the R23R, R23L : Although the authors tried to perform some bioinformatic analysis to inform the secondary structure of their synthesized peptides, I think it would be nice to add some CD experiments to prove the same. Computational analysis can be some what misleading and addition of CD experiments can be finally complemented with these algorithmic analysis.

Response: We fully agree with this proposal; however, it is impossible to measure the CD spectra of the R23R, R23L, R23R*, and R23L* peptides using our equipment under the studied buffer conditions. The problem is associated with the high level of noise generated by DMSO, which is present in the test solutions at a concentration of 2-20% (volume/volume).

4) Also please explain the results clearly as observed from Table 2.

Response: Additional text has been added after Table 2 to explain the results.

5) Experimental Cheking of the Antibacterial Activity of Peptides : For the agar plate experiments no negative control was used. I think the authors can simply use a scrambled peptide sequence or any peptide sequence which does not have any amyloidogenic properties as negative control. Also, they failed to compare the AMP effect of their synthesized peptides with already tested AMP against P. aeruginosa, for e.g. peptide ZY4 (Mwangi et al, PNAS, Dec 2019, 116 (52) 26516-26522; DOI:10.1073/pnas.1909585117) and Mel4 (Yasier et al, Scientific reports, 9, 7063 (2019). https://doi.org/10.1038/s41598-019-42440-2). I wonder how the test AMP in this study would have fared against the already known AMPs.

Response: We are grateful to the Reviewer for valuable comments and useful links to papers on the topic of the manuscript, which we reviewed and discussed in the context of our results. We fully agree that the use of a peptide lacking amyloidogenic sequences as a negative control is important. Moreover, we carried out similar experiments with the sequence RKKRRQRRRGG-Sar (R12Sar), which does not possess amyloidogenic properties. In the revised version of the manuscript, we added the Supplementary material demonstrating the absence of antimicrobial activity for R12Sar. However, it should be noted that this work already implied the use of the R23R peptide containing the LHITDMAWKR sequence, which did not exhibit amyloid-like properties during the experiments, as a kind of control sample. Thus, within the framework of this work, we have identified the peptide with the most pronounced antimicrobial properties among the synthesized R23R and R23L, R23R* and R23L* peptides. This peptide turned out to be the R23L* peptide containing the amyloidogenic sequence. R23L* has the greatest antimicrobial effect against ATCC28753 strain of P. aeruginosa in a liquid medium. We fully agree that in the future it is important to compare the antimicrobial effects of our hybrid peptide containing the amyloidogenic sequence not only with the known antibiotic (gentamicin sulfate), but also with the already known antimicrobial peptide that does not contain amyloidogenic sequences.

6) Measurement of the Antibacterial Activity of Peptides by Microdilution Technique: Figure 3:   Optical density decreases with decrease in the AMP concentration. Does this means test AMPs are more potent at lower concentration?

Response: This is a good question. Figure 3 was used to assess the intrinsic absorption of peptides. The data presented on it for high concentrations of peptides, of course, cannot be interpreted for tests of low concentrations of peptides on bacteria, so we did not use it in the revised version of the manuscript.

7) Toxicity of R23R and R23L: Why doesn't the authors included R23R and R23L in toxicity study especially when they have established that R23L* have more potent AMP activity as compared to others. Also, what is the premise to choose 8uM concentration for toxicity study.

Response: We have added additional experiments with peptides R23L* and R23R* on fibroblasts and lung carcinoma cells. The choice of the maximum concentration of 8 μM is due to the peculiarities of preparing concentrated solutions of peptides by dissolving in 100% DMSO, since peptides are poorly soluble in other solvents. Since DMSO at concentrations of 0.25% and higher is toxic to mammalian cells, we use multiple dilutions of the original peptide solutions. Due to the limited possibilities of dissolution of peptides, the final concentrations of peptides are 8 µM and lower, but it is possible to reduce the final concentration of DMSO so that it does not exceed 0.1%, and at such a concentration DMSO does not in itself have a toxic effect on eukaryotic cells. Thus, for peptide concentrations of 8 µM, we can test their toxic effect without the influence of DMSO.

8) Discussion: Please refer to any table or figures in the result section when you discuss about the results. Also, please explain the observation more clearly and state the advantages of these current AMPs over the already studied and published AMPs.

Response: Thank for the comment. Corresponding corrections have been made to the text of revised manuscript.

Reviewer 2 Report

In this manuscript, Grishin and colleagues demonstrate the antimicrobial effects of two peptides that were derived computationally and developed from sequences with different propensities for amyloid formation. Among the two peptides, only one showed antibacterial activity comparable with Gentamicin. The peptides were also not toxic to eukaryotes. 

This work is interesting and I have few comments and suggestions before it can be fully accepted:

1. English should be improved in many portions of the manuscripts. Also noticed several typos in the text.
The following are not exhaustive and authors are required to check their English in other parts of the manuscript.

line 26 - should be Gram-negative, not Gramm-negative
line 33 - This is misleading. you claimed "antibiotics" where you in fact you only tested Gentamicin. Mention only gentamicin.
line 65 - improve this sentence; unclear, missing verb - "inactivate once effective antibiotics, making them practically useless, for example"
Line 97 - Improve Figure 1 legend. Figure legends should be able to stand alone. Readers should be able to understand them without referring to the text. Add description of the peptides. What does underlined text mean. What about the red font residues, and many others?
line 113 - "In accordance with to", remove "to"
line 112 - "In this paperб", remove "6"
line 175 - Use "experimental validation" instead of "experimental checking"
Line 178 - should be Figure 2, not Figures 2
Line 269 - improve this sentence, unclear
Line 283 - improve the phrase - and does not affect its length for R23L*. Maybe use - but has no effect for peptide R23L*
Line 331 - "...growth condition-." remove "-"

2. What is the significance of figure 3? You presented it but never mentioned in discussion or elsewhere in the manuscript. Can you subtract this background from the signal found in Figures 4 and 5?

Author Response

In this manuscript, Grishin and colleagues demonstrate the antimicrobial effects of two peptides that were derived computationally and developed from sequences with different propensities for amyloid formation. Among the two peptides, only one showed antibacterial activity comparable with Gentamicin. The peptides were also not toxic to eukaryotes.

This work is interesting and I have few comments and suggestions before it can be fully accepted:

  1. English should be improved in many portions of the manuscripts. Also noticed several typos in the text.

The following are not exhaustive and authors are required to check their English in other parts of the manuscript.

line 26 - should be Gram-negative, not Gramm-negative

line 33 - This is misleading. you claimed "antibiotics" where you in fact you only tested Gentamicin. Mention only gentamicin.

line 65 - improve this sentence; unclear, missing verb - "inactivate once effective antibiotics, making them practically useless, for example"

Line 97 - Improve Figure 1 legend. Figure legends should be able to stand alone. Readers should be able to understand them without referring to the text. Add description of the peptides. What does underlined text mean. What about the red font residues, and many others?

line 113 - "In accordance with to", remove "to"

line 112 - "In this paperб", remove "6"

line 175 - Use "experimental validation" instead of "experimental checking"

Line 178 - should be Figure 2, not Figures 2

Line 269 - improve this sentence, unclear

Line 283 - improve the phrase - and does not affect its length for R23L*. Maybe use - but has no effect for peptide R23L*

Line 331 - "...growth condition-." remove "-"

 Response: We thank the Reviewer for his attention to our work and valuable comments. All suggestions have been taken into account by us and the corrections have been made in the text. English have been checked and corrected.

  1. What is the significance of figure 3? You presented it but never mentioned in discussion or elsewhere in the manuscript. Can you subtract this background from the signal found in Figures 4 and 5?

 Response: We agree that Figure 3 is not essential for demonstrating the antibacterial activity of the peptides. Figure 3 has been removed in the revised version of the manuscript; the numbers of other figures have been corrected.

Reviewer 3 Report

The current manuscript ‘Is it Possible to Create Antimicrobial Peptides Based on the Amyloidogenic Sequence of Ribosomal S1 Protein of P. aeruginosa?’ presents the use of amyloidogenic peptide sequences in forming novel antimicrobial peptides.  The  authors tried to address the effect of incorporating the antimicrobial properties of the cell penetrating peptide sequence linked to the amyloidogenic peptides. In multiple instances the authors failed to convey the necessity and advantages for creating such antimicrobial peptides (AMPs). Though the combinations of different cell penetrating peptides with amyloidogenesis peptides were made all of them showed minimal or no antibacterial effect except R23L* which showed antibacterial effect specifically in ATCC28753 strain of P. aeruginosa. Here are my comments for the manuscript.

The authors should introduce detailed discussion in the introduction with suitable literature on the amyloidogenesis in bacterial cells.

In line 101, the sentence states that the amyloidogenesis proteins can be toxic to the cells. What cells are being mentioned in this context? Is it bacterial cells or the eukaryotic cells? If they are eukaryotic cells then the line in 108 is a repetition of the sentence. The authors need to make the corrections accordingly.

The terms in table 1 and table 2 (Such as µHn, Hn,Z, I0, RF, SVM,ANN, DM..) should be explained as the footnotes of respective tables.

In line 213, the authors mentioned that the antimicrobial activity was tested at 20% DMSO. This seams to be so aggressive amounts of DMSO. Though it may not be reflected as solvent control, it could still increase the susceptibility of the AMPs. The authors can test the effect of the 20% DMSO on the bacterial susceptibility to the antibiotic (gentamicin sulfate) to see if the activity of the antibiotic increases in presence of DMSO or not. The data gentamicin, gentamicin+20% DMSO should also be incorporated in the figure 3 and figure 4.

In figure 4B, the corresponding data points for the positive control gentamycin at high concentrations in the graph should be designated with a special character such as * or # to signify that the optical density is almost or equal to zero and the same be mentioned in the footnotes to the graph. Currently it appears as if the data is missing in the graph.

In line 235 the authors claim that the R23R* is bacteriostatic at 12 µM correlating figure 5A. However, the optical density of the respective concentration is similar to the control (control shows O.D of 0.3 while R23R* at 12 µM is slightly above 0.25). The molecules theoretically can’t be considered to have any antibacterial effect with such minor changes in the optical densities. How did the authors conclude that its bacteriostatic with such a low difference in the bacterial growth? Moreover, in figure 5b, the error bars for the activity of R23* is also too high. Still the authors claim that its I effective antibacterial compound. Did they investigate if its bactericidal or bacteriostatic?

I read the whole discussion thrice and it is still unclear to me why the authors have chosen amyloidogenic peptide sequences of P.aeruginosa for creating the AMPs and what advantages did it give over the normal antibiotics. No studies pertaining to prevention of antibiotic resistance were performed too.

Overall, though the authors have made efforts to produce AMPs using the amyloidogenic proteins, they failed to convey their hypothesis due to lack of proper experimentation as well as poor data presentation as well as its interpretation. This manuscript could not be accepted for publication in its current form. I therefore suggest the authors to revise and try resubmitting it.

Author Response

The authors should introduce detailed discussion in the introduction with suitable literature on the amyloidogenesis in bacterial cells.

 Response: Thanks to the Reviewer for the valuable comments. A discussion of studies of amyloidogenesis in bacterial cells has been added to the Introduction section.

In line 101, the sentence states that the amyloidogenesis proteins can be toxic to the cells. What cells are being mentioned in this context? Is it bacterial cells or the eukaryotic cells? If they are eukaryotic cells then the line in 108 is a repetition of the sentence. The authors need to make the corrections accordingly.

The terms in table 1 and table 2 (Such as µHn, Hn,Z, I0, RF, SVM,ANN, DM..) should be explained as the footnotes of respective tables.

 Response: The recommended corrections have been made.

In line 213, the authors mentioned that the antimicrobial activity was tested at 20% DMSO. This seams to be so aggressive amounts of DMSO. Though it may not be reflected as solvent control, it could still increase the susceptibility of the AMPs. The authors can test the effect of the 20% DMSO on the bacterial susceptibility to the antibiotic (gentamicin sulfate) to see if the activity of the antibiotic increases in presence of DMSO or not. The data gentamicin, gentamicin+20% DMSO should also be incorporated in the figure 3 and figure 4.

 Response: This remark is not entirely correct, since the optical density of peptides without cells was tested in 20% DMSO, and not the antimicrobial properties of peptides. To avoid further confusion, and in accordance with the comments of Reviewers 2 and 3, we have deleted Figure 3 and clarified the figure captions, which indicated that the peptide and gentamicin preparations contained a final DMSO concentration of 2%. Thus, all preparations and controls contained the same final concentration of DMSO. Thus, the influence of this solvent on the antimicrobial effects of peptides was leveled out.

In figure 4B, the corresponding data points for the positive control gentamycin at high concentrations in the graph should be designated with a special character such as * or # to signify that the optical density is almost or equal to zero and the same be mentioned in the footnotes to the graph. Currently it appears as if the data is missing in the graph.

 Response: Thanks for the valuable comment. Corresponding corrections have been done.

In line 235 the authors claim that the R23R* is bacteriostatic at 12 µM correlating figure 5A. However, the optical density of the respective concentration is similar to the control (control shows O.D of 0.3 while R23R* at 12 µM is slightly above 0.25). The molecules theoretically can’t be considered to have any antibacterial effect with such minor changes in the optical densities. How did the authors conclude that its bacteriostatic with such a low difference in the bacterial growth? Moreover, in figure 5b, the error bars for the activity of R23* is also too high. Still the authors claim that its I effective antibacterial compound. Did they investigate if its bactericidal or bacteriostatic?

 Response: We completely agree with the remark and have removed the inaccurate statement that arose during the preparation of the manuscript.

I read the whole discussion thrice and it is still unclear to me why the authors have chosen amyloidogenic peptide sequences of P.aeruginosa for creating the AMPs and what advantages did it give over the normal antibiotics. No studies pertaining to prevention of antibiotic resistance were performed too.

Overall, though the authors have made efforts to produce AMPs using the amyloidogenic proteins, they failed to convey their hypothesis due to lack of proper experimentation as well as poor data presentation as well as its interpretation. This manuscript could not be accepted for publication in its current form. I therefore suggest the authors to revise and try resubmitting it.

 Response: Undoubtedly, the use of amyloidogenic peptides for the development of AMPs is of interest within the framework of the hypothesis of targeted coaggregation with the target protein. In general, the creation of hybrid peptides containing a CPP sequence and an amyloidogenic region allows several mechanisms of the antimicrobial action of peptides to be exploited. On the one hand, due to the penetration into the cell and destruction of the bacterial cell membrane, and on the other, directed coaggregation with a functionally important vital bacterial protein. Based on the results of the development of these peptides, we assume that bacterial resistance to AMPs, acting through several mechanisms, will develop much longer than to conventional antibiotics, if at all. Of course, our assumptions regarding several mechanisms, as well as the long-term action of peptides, are planned to be studied in the future. A great achievement of this work is that even among the first peptides synthesized by us based on the S1 sequence from P. aeruginosa, encouraging results were obtained for the R23L* peptide, that is, antimicrobial effects comparable to the action of the antibiotic gentamicin sulfate. The corresponding comments have been added to the text of the manuscript.

Round 2

Reviewer 1 Report

The manuscript entitled " Is it possible to create antimicrobial peptides based on the amyloidogenic sequence of ribosomal S1 protein of P. aeruginosa?" by Grishin et al describe the use of novel hybrid amyloidogenic peptide as novel antimicrobial therapeutics against P. aeruginosa. The authors had taken the inputs from the reviewer and have tried to  substantially improved the whole manuscript. The revised manuscript now have more logical flow with results now clearly stated. However, the manuscript still have some following moderate flaws which need to be addressed before it is ready for publications. 

1) Line 51  -  "Such Server as"  can be replaced by Webservers or Online programs

2) Line 67-69  - Simplify the whole statement as it is still not clear enough.

3) Line 68- Replace "of" with "within"

4) Line 81- This is redundant and should be merged with the following statements.

5) Line 95- "it is found only win bacteria" replace the sixth " and is found as"

6) Line 101- " number of functionalities". Does this means each domain have unique functions. If yes, please mention them briefly, specially the domain with amyloidogenic property.

7) Line 102- "represent" replace this with "Consist of an"

8) Line 102-104 "B-barrel assembled from a-helix" Please reframe this whole sentence. In original it is misleading.

9) Line 106-107- Clarify or reframe this whole statement. Which S1 domains and the repeats authors are referring to?

10) Line 119- Mention the domain number from which these peptide sequence was taken for the present study.

11) Line 125- "based on the amyloidogenic peptide" This is also redundant. Can be removed.

12) Line 174- "Residues similar" Replace this with " residues which are similar"

13) Line 186- " the introduction of a". This can be removed

14) Line 186- "at" replace this with "of"

15) Line 187-189- Explain the changes observed on the secondary structure upon sarcosine substitution

16) Line 195-197 Consider reframing this whole sentence for better understanding.

17) Line 215-216 Simply the sentence

18) Line 217- "are" replace this with "does"

19) Does the Figure 3( from original version ) is being removed from the manuscript? It is bit confusing as the figure remains but the legend is striked off.

20) Line 264-265- Simplify this whole statement. It is hard to understand what the authors are implying here.

21) Line 370-371- Elaborate or briefly discuss the Z and Io obtained values contributing to anti-microbial properties.

22) Line 418- This is contradictory and need to be reframed as in the introduction and in figure legend 1, it is mentioned that R23R and R23R* did contain amyloidogenic sequence, however they failed to identify amyloidogenic properties through their conducted studies.  

23) Line 455-456 Please clarify the statement what actually authors want to convey here. Does they mean bacteria will develop resistance against R23L* or ZY4 or Mel4.

24) Line 583 Please provide brief description about the methodology of  cytotoxicity assay

Author Response

Thanks to the Reviewer for the valuable comments. The recommended corrections have been made.

Reviewer 3 Report

The authors have made all the suggested modifications as well as corrections to the manuscript. I feel that the manuscript can be accepted for publication.

Author Response

Thanks to the reviewer for a positive assessment of our article.